# A High Proportion of Malaria Vector Biting and Resting Indoors despite Extensive LLIN Coverage in Côte d’Ivoire

**DOI:** 10.3390/insects14090758

**Published:** 2023-09-12

**Authors:** Naminata Tondossama, Chiara Virgillito, Zanakoungo Ibrahima Coulibaly, Verena Pichler, Ibrahima Dia, Alessandra della Torre, Andre Offianan Touré, Akré Maurice Adja, Beniamino Caputo

**Affiliations:** 1Entomology and Herpetology Unit, Institut Pasteur de Côte d’Ivoire, Abidjan 01 PB 490, Côte d’Ivoire; amitondo@gmail.com (N.T.); zana_ib@yahoo.fr (Z.I.C.); 2Laboratoire de Biologie et Santé, UFR Biosciences, Université Félix Houphouët Boigny Cocody, Abidjan 01 BP V34, Côte d’Ivoire; adjamaurice@yahoo.fr; 3Department of Public Health and Infectious Diseases, Institute Pasteur Italia-Fondazione Cenci-Bolognetti, University of Rome ‘Sapienza’, Piazzale Aldo Moro 5, 00185 Rome, Italy; chiara.virgillito@uniroma1.it (C.V.);; 4Pôle de Zoologie Médicale, Institut Pasteur de Dakar, 36 Avenue Pasteur, Dakar BP 220, Senegal; ibrahima.dia@pasteur.sn; 5Unité de Paludologie, Institut Pasteur de Côte d’Ivoire, Abidjan 01 PB 490, Côte d’Ivoire; andre_offianan@yahoo.fr; 6Institut Pierre Richet, Institut National de Santé Publique, Bouaké 01 BP 1500, Côte d’Ivoire

**Keywords:** *Anopheles gambiae*, malaria transmission, bednet, *Plasmodium falciparum*, West Africa

## Abstract

**Simple Summary:**

Malaria—a major cause of mortality in Africa—is transmitted to humans by Anopheles mosquitoes. The main approach to prevent bites and malaria transmission is to sleep under bed-nets impregnated by insecticides, which are largely available to the population. We collected mosquitoes either found indoors in the morning, or exiting the houses during the night, as well as information on the use of bed-nets in the sample houses, to help understanding the impact of these approach on mosquito biting and resting behavior. We confirmed high levels of malaria transmission in the two villages studied, despite a high use of bed-nets. A large proportion of Anopheles females exiting the houses during the night were not blood-fed, suggesting that bed-net use prevented them to find an available host indoors and led them to move outdoors to find one. However, a large proportion of females collected indoors were blood-fed, suggesting that prevention of bites was incomplete even when all people in a house declared to have slept under bed-net. Overall, results suggest a major role of bed-nets in protecting from mosquito bites and malaria transmission, but highlight that the level of protection is limited, likely due to mosquito and human behaviors.

**Abstract:**

Malaria is still a leading cause mortality in Côte d’Ivoire despite extensive LLINs coverage. We present the results of an entomological survey conducted in a coastal and in an inland village with the aim to estimate *Anopheles gambiae* sensu lato (s.l.) female’s abundance indoor/outdoor and *Plasmodium falciparum* infection rate and analyze the occurrence of blood-feeding in relation to LLINs use. Pyrethrum spray (PSC) and window exit traps (WT) collections were carried out to target endophagic/endophilic and endophagic/exophilic females, respectively. Data on LLINs use in sampled houses were collected. (1) high levels of malaria transmission despite LLINs coverage >70% (~1 *An. gambiae* s.l. predicted mean/person/night and ~5% *Plasmodium falciparum* infection rate); (2) 46% of females in the PSC sample were blood-fed, suggesting that they fed on an unprotected host inside the house; (3) 81% of females in WT were unfed, suggesting that they were leaving the house to find an available host. Model estimates that if everyone sleeps under LLINs the probability for a mosquito to bite decreases of 48% and 95% in the coastal and inland village, respectively. The results show a high proportion of mosquito biting and resting indoors despite extensive LLINs. The biological/epidemiological determinants of accounting for these results merit deeper investigations.

## 1. Introduction

Malaria remains the main reason for consultation at health facilities and a leading cause of morbidity and mortality in Côte d’Ivoire, as well as in the neighbouring countries of the West African sub-region, where in 2021 it caused ~120 million human cases and ~328.000 deaths [1]. One of the reasons for this very high malaria burden in West Africa is the presence of very efficient vector species belonging to the *Anopheles gambiae* (Diptera: Culicidae) complex (hereafter *An. gambiae* s.l.), in particular the two most synanthropic members, *An. gambiae* s.s. and *An. coluzzii* Coetzee & Wilkerson, 2013 [2]. In Côte d’Ivoire, the species’ abundance follows different seasonal patterns in different regions, e.g., with peaks in May and October in southeastern and central regions, respectively (Figure 1A) [3].

Distribution of long-lasting insecticidal nets (LLINs) and indoor residual spraying (IRS) are the main interventions implemented to reduce human-vector contact and vector abundance, i.e., to protect the human host from mosquito bites and reduce the intensity of local transmission. In Côte d’Ivoire, IRS was not largely implemented until 2020, while three mass LLIN distribution campaigns were implemented in 2011, 2014–2015 and 2017–2018. Thanks to these campaigns, LLINs coverage increased from 10% in 2006 to 67% in 2012 and 76% in 2016 [4]. The proportion of population that use LLINs increase from 62% in 2016 to 75% in 2019 (data obtained post-campaign in 2017–2018) [4,5]. Despite an initial decline in malaria transmission, an increase by 10.4% of the estimated number of confirmed malaria cases was recorded between 2017 and 2020 (from 260 to 287 cases per 1000 population) [1]. The same pattern is observed in other neighbouring West-African countries. Many factors have been implicated in reducing LLIN efficacy at the community level: (i) household practises such as poor maintenance (e.g., LLINs with holes [6,7] loss of physical integrity [7,8,9] frequent washing, excessive number of people sleeping under the bed net [10] and infrequent use of LLINs [11], (ii) human factors such as sleeping time, night-time activities and time spent outdoors at night [12,13,14] and (iii) resistance to pyrethroid insecticides used to impregnate the nets in the vector populations [15,16,17]. Physiological resistance mechanisms are widespread in *An. gambiae* s.l.; coastal populations from Côte d’Ivoire show high phenotypic resistance to all classes of insecticides available for malaria control [18], while inland populations are characterised by high frequencies of pyrethroid resistance *kdr* alleles (e.g., L1014F) [19,20]. Finally, vector species have also developed behavioural adaptations to malaria control practices, such as shifting of biting behaviour in time (daytime biting inside human dwellings [21] and space (biting outside human dwellings [22,23,24]. In Côte d’Ivoire, a higher proportion of *An. gambiae* s.l. biting outdoors was reported in the coastal area compared to inland [16].

Here we present the results of an entomological survey conducted in 2018–2019 in a coastal village and an inland village in Côte d’Ivoire with the aim to estimate (1) relative density and *Plasmodium falciparum* infection rate of *An. gambiae* s.l. females resting indoors and leaving houses at night (LM-1 and GLMM-2); (2) blood- feeding status of the *An.gambiae* s.l. resting indoors and leaving houses at night (GLMM-3); (3) the occurrence of indoor blood-feeding in relation to LLIN use (GLMM-4).

## 2. Materials and Methods

### 2.1. Study Sites

The study was carried out in two villages of Côte d’Ivoire selected as sentinel sites by the National Malaria Control Program (NMCP): Piste 4 in the southeastern region of Aboisso, 100 km east of the capital city Abidjan and 50 km from the coast (5°28′ N and 3°12′ W; hereafter coastal village) and Pétessou in the central region 11 km south from the city of Bouaké and >250 km from the coast (8°6′ N and 5°28′ W; hereafter inland village) (Figure 1A).

The coastal village has an estimated population of <300 inhabitants living in houses built of banco (a mix of mud and straw). The climate is hot, with two rainy seasons (March–June and September–October) and two dry seasons (July–August and December–February). The inland village has an estimated population of 700 inhabitants living in houses built of hard bricks. The climate is tropical and humid, with a rainy season from April to October and a dry one from November to March.

LLIN (Dawa plus2.0, treated with deltamethrin, Permanet 2.0 treated with deltamethrin MagNet treated with alpha-cypermethrin, Yorkool treated with deltamethrin and Olyset treated with permethrin) were distributed in 2017 reaching a coverage rate of 95.8% and 97.2% in the coastal and inland region, respectively [5].

### 2.2. Mosquito Sampling and Field Processing

Mosquitoes were sampled in the two above villages in December 2018 and in March, May and October 2019. Pyrethrum spray collections (PSC), to collect mosquitoes with endophilic behaviour [25], i.e., mosquitoes which tend to inhabit/stay indoors (https://www.cdc.gov/malaria/glossary.html#, accessed on 5 June 2023), and by window exit traps (WT), to collect mosquitoes with exophilic behaviour [25], i.e., mosquitoes that tend to inhabit/stay outdoors (https://www.cdc.gov/malaria/glossary.html#, accessed on 5 June 2023) see (Appendix A). PSCs were carried out daily between 6 and 8 AM in randomly selected houses/village house were sampled once over the entire sampling period. All exposed surfaces in each house were covered with white sheets before spraying the rooms with pyrethrum (according to the standard [26] protocol), to allow for the collection of all dead mosquitoes afterwards. Window exit trap collections were implemented in five houses/village (different from those selected for PSCs); each house was sampled for 4 consecutive nights, and specimens inside WT were collected early in the morning. After each sampling, the house householder head was asked about the number of people spending the night in the sampling house and the number of people sleeping under the net, and the replies were recorded. Ethical approval for the study was granted by Ministère de la Santé et de l’Hygiene Publique (Comité National D’Ethique de la Recherche, CNER, 023–18/1VISHP/CNER, Abidjan 3 April 2018 ). Free and informed consent for collection was obtained from the household heads. Collected mosquitoes were identified using the Gilles and De Meillon keys [27] for Anopheles fauna and Detinova keys [28] for abdominal status (unfed, freshly fed, semi-gravid and gravid). *Anopheles gambiae* s.l. specimens were kept dry in micro-tubes containing silica gel for molecular analyses.

### 2.3. Molecular Analyses

*Anopheles gambiae* s.l. females were dissected into two parts, i.e., head-thorax and abdomen. DNA was extracted using the DNAzol reagent (MRC. Inc., Cincinnati, Ohio) according to the protocol described by Rider et al. [29]. DNA extracted from the head-thorax was used as a template for identification of *Anopheles gambiae* complex species using the ribosomal DNA PCR-based assay described by Wilkins et al. [30] and *Plasmodium falciparum* detection by the Real-Time-PCR-based assay described by Bass et al. [31]. DNA extracted from the abdomen of blood-fed females was used to identify the origin of blood meal following the procedure described by Kent et al. [32].

### 2.4. Statistical Analysis

*Anopheles gambiae* relative density: To evaluate the mean *An. gambiae* s.l. females per person per night (m/p/n) in the coastal and inland villages as a function of sampling methods (PSC and WT), a regression analysis was performed using the following covariates: villages (coastal and inland) and sampling methods (PSC and WT). The response variable Y was defined as the number of *An. gambiae* s.l. divided by the number of people sleeping in the room on the day of sampling [33]. The response variable was modelled by Gaussian distribution with mean µ and standard deviation σ. The two covariates (villages, sampling methods) and their interaction were included in the model to ascertain whether the abundance (m/p/n) in the inland and coastal villages was influenced by the sampling methods, and whether there was a difference between the two villages for the same sampling methods. Both linear models (LM-1) and linear mixed effect models (LMMs-1) were tested, considering houses as a “random effect” in the intercept to account for individual variability. Akaike information criteria (AIC) were used to compare the models to decide whether to include the random effect.

*Plasmodium falciparum* infection rate (P*f*R) of *An. gambiae* s.l. females was estimated by regression analysis. Assuming that the response variable (i.e., presence and absence of *P. falciparum* DNA in the head + thorax of processed specimens) follows a binomial distribution, P*f*Rwas modelled as a function of villages and sampling methods. These two covariates and their interaction were included in the model to ascertain whether the probability of finding *P. falciparum* was influenced by the sampling method and whether there was a difference between the two villages. Both generalized linear models (GLM-2) and generalized mixed effect models (GLMMs-2) were tested, considering houses as a “random effect” in the intercept to account for individual variability. Akaike information criteria (AIC) were used to compare the models to decide whether to include the random effect.

Blood-feeding status in relation to sampling methods. The probability of finding blood-fed *An. gambiae* s.l. in PSC vs WT collections was estimated by GLMM-3. A binomial distribution was used, assuming the response variable “blood-fed” (i.e., sum of freshly fed and half gravid females) as a success, and “empty” (i.e., sum of unfed and gravid) as a failure, and was modelled as a function of the sampling methods. The random structure (i.e., house of collection) was included in the model a priori. The model was run separately for the two villages.

Blood-feeding status in relation to use of LLINs. The probability of finding blood-fed *An. gambiae* s.l. as a function of LLIN use was estimated by GLMM-4. A dichotomous variable “use of LLINs” was defined as 0 if nobody slept out of the LLIN, and 1 if everyone slept under a LLIN. A binomial distribution was used, assuming the response variable “blood-fed” (i.e., sum of freshly-fed and half-gravid females) as a success, and “empty” (i.e., sum of unfed and gravid females) as a failure. This model was run including collections in both villages either by WT and PSC or by PSC only. In addition, an interaction term between “use of LLINs” and village was considered to account for different patterns of probability of finding blood-fed females in the two villages due to different use of LLINs.

All analyses were conducted with the R version 4.2.3 [34].

## 3. Results

A total of 617 *Anopheles* females belonging to five species were collected from the coastal (*n* = 275) and inland (*n* = 368) villages (Appendix A). Overall, *An. gambiae* s.l. accounts for 97.7% and 82.9% of the total captures in the coastal and inland village, respectively. *Anopheles funestus* is found only inland (17.1%), and *Anopheles nili* (1%), *Anopheles coustani* (1%) and *Anopheles salbaii* (1%) are found only in the coastal village. The following analyses were carried out only on *An. gambiae* s.l., due to its high prevalence in both villages.

### 3.1. Anopheles gambiae Relative Density and Plasmodium falciparum Infection Rate

A total of 269 and 308 *An. gambiae* s.l. females were collected in the coastal and inland village, respectively (Appendix A). The results of model selection, developed to estimate *An. gambiae* s.l. per person per night (m/p/n) as a function of the sampling method, show a lower AIC for LM-1 (AIC = 1259.01) than for LMM-1 (AIC = 1264.27), indicating that there is no need to include collection houses as a random effect. No significant interaction term was found in LM-1, indicating no statistical difference between the estimates of m/p/n in the two villages sampled using either PSC (coastal: 1.24, 95%CI 0.63/1.85; inland: 1.08, 95%CI 0.60/1.56) or WT (coastal= 0.76, 95%CI 0.29/1.24; inland = 0.50, 95%CI 0.03/0.97) (Table 1 and Appendix A). Furthermore, the estimates of *An. gambiae* s.l. per person per night based on either PSC or WT were comparable in both villages, i.e., no statistical difference was observed between the two sampling methods in both villages (Table 1 and Appendix A).

*Plasmodium falciparum* infection rate (P*f*R) was 11% (24 positive *An. gambiae* s.l. females out of 207 successfully analysed) and 17% (41/238) in the coastal and inland villages, respectively. Results of model selection, developed to estimate P*f*R in the two villages as a function of the sampling method, showed lower AIC for GLM-2 (AIC = 174.63) than for GLMM-2 (AIC = 173.96), indicating no need to include collection houses as random effect. No significant interaction term was found in GLM-2, indicating no statistical differences in P*f*R estimates between the two villages considering the same sampling methods (Table 1 and Appendix A). However, in both villages the estimates of P*f*R in samples collected by PSC was statistically higher than those estimated in samples collected by WT (*p*-value < 0.001).

### 3.2. Anopheles gambiae Human Blood Index (HBI) and Physiological Status and Association with Bed-Net Usage

The Human Blood Index (HBI) was 98% in the coastal village (*n* = 50) and 100% in the inland village (*n* = 41) village (Appendix A); only one engorged mosquito from the coastal village (sampled by WT) did not feed on humans, but on bovine/goats.

Of the 556 females examined for physiological status, 53% were “empty” (i.e., 218 unfed and 77 gravid) and 47% blood-fed (i.e., 109 freshly fed and 152 half-gravid) (Figure 1B; Table 2). Empty females predominated in WT collections (coastal village = 65%; inland village = 93%), whereas blood-fed females predominated in PSC collections (coastal village = 75%; inland village = 67%). GLMM-3 results show that in both villages the estimated proportion of blood-fed females is lower in WT collections (coastal = 24.2%, 95%CI 9–50%; inland = 5.4%, 95% CI 1–16%) compared to PSC (coastal = 87.3%; 95%CI 69–95%; inland = 71.1%; 95%CI 52–85%; *p*-value < 0.005) (Appendix A).

Overall, the proportion of blood-fed females was 57% in the coastal village and 37% in the inland village. The proportion of sampled houses provided with LLINs (i.e., coverage) throughout the sampling period was 75% (*n* = 40) and 86% (*n*= 25) in the coastal and inland village respectively. The observed mean proportion of people sleeping in the house under LLINs (i.e., usage) was 71.0% and 89.7% in the coastal and inland village, respectively (Appendix A). GLMM-4 shows that in both villages, the probability of finding a blood-fed female was significantly lower when everyone slept under the LLINs (coastal = 48%, 95%CI 40–56%; inland = 27%, 95%CI 22–33%) than when nobody does (coastal = 66%, 95%CI 57–77%; inland = 87%, 95%CI 74–94%; Appendix A). The results of GLMM-4 shown that if everyone sleeps under LLINs the probability to find a blood-fed female decreases of 48% (95%CI 18–72%) and 95% (95%CI 87–98%) in the coastal and inland village, respectively. In addition, focusing on PSC collections, the probability to find a blood-fed female was statistically lower in houses where 100% of the people were sleeping under LLIN only in the inland village (*p* < 0.001; Appendix A).

### 3.3. Anopheles gambiae s.l. Species Composition

Of the 527 *An. gambiae* s.l. specimens successfully genotyped for diagnostic SNPs in the IGS-rDNA region, 43.1% were *An. gambiae* s.s., 34.9% were *An. coluzzii* and 22% showed a heterozygous IGS pattern (hereafter, IGS-hybrids). *Anopheles coluzzii* and *An. gambiae* s.s. were found throughout the sampling season in both villages (Figure 2a). IGS-hybrids were found at frequencies ranging from 55.6% in December 2018 and 7.7% in October in the coastal village (*n* = 99) and were rare in the inland (*n* = 17) (Figure 2a). No significant differences in species composition were observed between the two sampling methods (Figure 2b) (coastal village: PSC = 33.8% *An. coluzzii*, 28.1%, *An. gambiae* s.s. and 38.1% IGS-hybrids; WT = 31.9% *An. coluzzii*, 28.5%, *An. gambiae* s.s. and 39.6% IGS-hybrids; $\chi^{2}=0.08$, *p*-value = 0.95; PSC = 42.7% *An. coluzzii*, 50.7%, *An. gambiae* s.s. and 6.6% IGS-hybrids; WT = 30.6% *An. coluzzii*, 63.5%, *An. gambiae* s.s. and 5.9% IGS-hybrids $\chi^{2}=3.45$, *p*-value = 0.17).

## 4. Discussion

Results show high level of malaria transmission in both sampled villages in Côte d’Ivoire despite extensive LLIN coverage, as well as a high proportion of *An. coluzzii* and *An. gambiae* (and of IGS-hybrids, particularly in the coastal village) biting and resting indoors.

The predicted number of *An. gambiae* s.l. female/person/night (~1) with both sampling methods is in the range of that estimated based on CDC trap collections in the same villages during the same period [22] and in the coastal village in 2021 [35]. These estimates may be an underestimation, as shown by >20 b/p/n estimates calculated based on human landing catches [19,20], which are known to have a higher collection efficiency than other sampling approaches. The estimated *Plasmodium falciparum* infection rates for females collected by WT (coastal = 5%, inland = 4%) are in the range of those obtained across the country in recent years [35]. Notably, however, P*f*R estimates for samples collected by PSC are significantly higher (coastal = 20%, inland = 35%), but not realistic due to the bias associated with the high sensitivity of the Real-Time-PCR approach, which has been shown to detect contamination of DNA extracted from thorax-head with blood containing *Plasmodium* gametocytes [36]. This is confirmed by the very high proportion of blood-fed females (71%) in the PSC samples and the significantly higher proportion of blood-fed (80%) compared to unfed females (20%) among *Plasmodium* positive ones (N = 65; $\chi^{2}$ = 32.12; *p*-value < 0.001).

By comparing conventional indoor PSC (targeting the fraction of the population resting indoors at night) and WT collections (targeting the fraction of the population leaving houses during the night), and by recording the number of people sleeping under LLIN in the sampled houses, we were able to analyse the behaviour of *An. gambiae* s.l. in relation to the presence of bed-nets. The results clearly show: (i) a comparable number of females resting indoors and leaving the houses at night; and (ii) a high level of blood-feeding indoors (46% of blood-fed females in PSC sample, *n* = 261/556) despite the proportion of people sleeping under LLIN (71% and 90% in the coastal and inland village, respectively) is well above 50%, i.e., the conventional minimum threshold required to achieve community protection [37,38,39,40]. This latter result is consistent with observation from Kenya where human landing catches collections revealed high number of Anopheles biting indoor despite high coverage of LLINs [41].

Most of the females collected resting indoors were freshly fed or half-gravid, suggesting that they fed on an unprotected host inside the house and that their post-prandial resting was not affected by LLINs. Non-negligible occurrence of blood-feeding despite the use of LLINs was observed also in experimental huts in Boukè region [42]. On the other hand, most of the females leaving the houses were unfed, suggesting that they could not find an available host indoors and were leaving the house for host-seeking, thus contributing to outdoor malaria residual transmission [43,44].

It would be interesting to understand if there is any biological difference between the two fractions of the population (the one resting indoors and that leaving the house) that could account for the indoor success of only one of them, or if this is a random behaviour. It can be hypothesized that the females biting and resting indoors are “early-biters” and had access to hosts before they go to sleep under bed-net, whereas the fraction of females leaving the house during the night were “late-biters”, unable to find an unprotected host indoors. A shift in biting time has been reported from several regions and is recognised as contributing to residual malaria transmission despite high LLIN coverage [21]. Alternatively, females biting and resting indoors may have a higher resistance to insecticides than those leaving the house. However, the existence of a fraction of the population that is still highly susceptible to and repelled by pyrethroids is not supported by data showing very high levels of resistance in *An. gambiae* s.l. in Côte d’Ivoire, nor by the frequencies >85% of *kdr* alleles reported from the two villages [22]. A third hypothesis is that the two fractions may have a different *An. gambiae* s.l. species composition and that different species may have different feeding and resting behaviours in the regions. The limited sample sizes in the present work do not allow to carry out the analyses at the species level, but we confirmed the sympatric presence of *An. coluzzii* and *An. gambiae* s.s. in both villages and a high proportion of IGS-hybrid specimens, especially in the coastal site, as already shown by [22]. Finally, also the human factors such as sleeping time and night-time activities could have an impact on the LLINs protection [12,13,14]. Further studies on larger samples will allow to investigate potential differences in feeding and resting behaviour between these three groups, whereas genomic analysis of IGS-hybrids is required to clarify the true nature of these specimens. Intriguingly, genomic data suggest that they may not be the result of current hybridization between the two species [45].

Finally, modelling results show that if everyone sleeps under LLINs the probability for a mosquito to bite decreases of 48% (95% CI 18–72%) and 95% (95% CI 87–98%) in the coastal and inland village respectively. These results are consistent with results from a study in Benin [46] showing a 66% reduction in the probability of being bitten by *An. gambiae* when sleeping under bed nets and with those from a small-scale field study carried out in the South Côte d’Ivoire showing that Permanet 2.0 bed net have a low blood-finding inhibition (~65%; [47]).

Further studies are needed to confirm the results obtained and understand whether the different protection by LLINs estimated between the two villages is due to differences in human behaviour before going to sleep under the bed-net, and/or in biting rhythms of mosquito vectors under the tropical and savannah conditions, and/or to differences in LLINs efficacy.

## 5. Conclusions

According to the [1] LLINs were the main drivers of the decline in malaria transmission and burden between 2005–2015, particularly in moderate-to-high transmission settings, but their effectiveness has been declining since 2015. The results here presented provide evidence that in coastal area of Côte d’Ivoire, where malaria incidence is increasing in recent years—sleeping under a LLIN does not fully prevent being bitten by a malaria vector, and thus malaria transmission, nor the possibility for Anopheles females to digest their blood-meal indoors. This may be due to high resistance to pyrethroids, which reduces both the protective and repellent effects of LLINs, as shown in several Afrotropical settings [48,49,50], but also to behavioural changes in vector populations, such as changes in biting times, exposing humans to indoor biting before sleeping under the net or immediately upon waking up [21]. Moreover, results reveal high behavioural plasticity of *An. gambiae* s.l. females, a large proportion of which enters the houses but leaves without feeding, presumably in search of hosts outdoors and contributing to outdoor malaria transmission. This may reflect a shift towards an exophagic/exophilic behaviour already reported from other Afrotropical regions [40,43,51]

## Figures and Tables

**Figure 1 insects-14-00758-f001:**
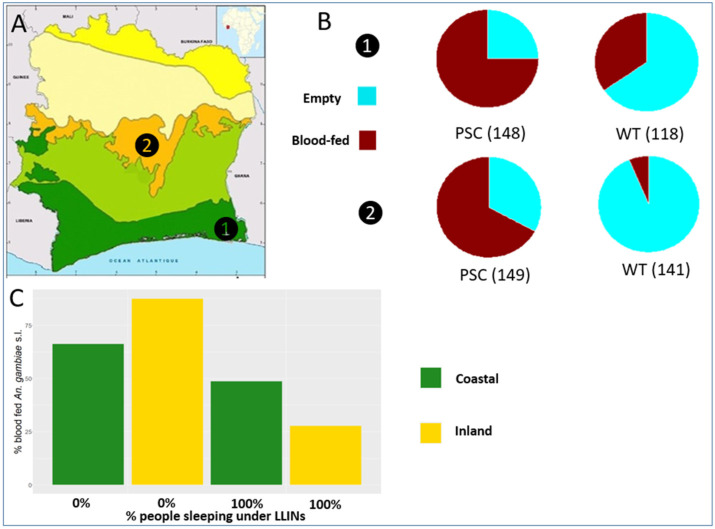
(**A**) Location of the two sampling villages in Côte d’Ivoire (1 = Piste 4; 2 = Petessou); (**B**) Frequencies of “empty” and blood-fed *An. gambiae* s.l. females collected by pyrethrum spray collections (PSC) and window exit traps (WT) in a coastal (1) and inland village (2). (**C**) Observed mean percentage of blood-fed females in the two villages as a function of LLIN use (0% nobody sleeping under LLINs100% everybody sleeping under LLINs).

**Figure 2 insects-14-00758-f002:**
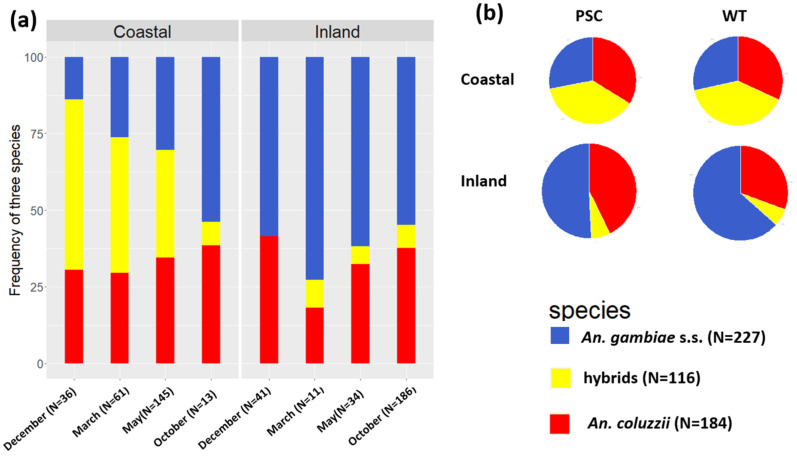
(**a**) Frequency of *An. coluzzii* (red) *An. gambiae s.s.* (blue) and IGS-hybrids (yellow) in 4 sampling periods in 2018–2019 in two villages from Cote D’Ivoire, and (**b**) in Pyrethrum Spray Collections (PSC) and Window Exit Traps (WT).

**Table 1 insects-14-00758-t001:** *Anopheles gambiae* s.l./person/night (m/p/n) and *Plasmodium falciparum* rate (P*f*R) in samples collected by Pyrethrum Spray Collections (PSC) and Window Exit Traps (WT) in a coastal and inland village in Cote d’Ivoire in 2018–2019, estimated by LM-1 (m/p/n) and GLM-2 (P*f*R). N = number of observations (collection houses) for each site and sampling method; M = number of females tested for *P. falciparum* infection (number of infected females).

Village	Sampling Methods	N	m/p/n (95%CI)	M	P*f*R (95%CI)
Coastal	PSC	48	1.24 (0.63/1.85)	89 (18)	20% (13–29%)
	WT	80	0.76 (0.29/1.24)	118 (6)	5% (2–10%)
Inland	PSC	78	1.08 (0.60/1.56)	97 (34)	35% (26–45%)
	WT	80	0.50 (0.03/0.97)	141 (7)	4% (2–10%)

**Table 2 insects-14-00758-t002:** A total number of *An. gambiae* s.l. females collected unfed, half gravid, freshly fed, and gravid in the coastal and inland village using PSC and WT.

		Physiological Status				
Village	Samplig Method	Unfed	Half Gravid	Freshly Fed	Gravid	Tot
Coastal	PSC	20%	45%	30%	5%	148
	WT	51%	18%	17%	14%	118
Inland	PSC	16%	39%	28%	17%	149
	WT	74%	4%	2%	20%	141

## Data Availability

Data are available online at https://github.com/Chia1992 (accessed on 5 June 2023).

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
