# Peer review of "A High Proportion of Malaria Vector Biting and Resting Indoors despite Extensive LLIN Coverage in Côte d’Ivoire"

_insects, 2023, doi:10.3390/insects14090758_

Round 1

Reviewer 1 Report

This study summarizes an evaluation of the vector mosquito contribution to malaria transmission in two villages of Côte d'Ivoire with different ecological contexts (coastal and inland).  The primary goal is to determine the extent of human blood feeding and parasite transmission despite the apparent extensive use of long lasting insecticidal nets (LLINs).  Overall, the study was carried out well and the manuscript summarizes the results adequately with good narrative, description, data summarization, and data analysis.  Although no problems with interpretation were found during review, the manuscript needs editing and revision of a minor nature.  The primary finding: that most vector contact appears to occur indoors even in the presence of LLINs, is important to report in the scientific literature to counter the notion that outdoor biting has replaced indoor biting.  This is decidedly not true as many studies have shown, yet the latter notion continues to receive credence as if it is now dogma; it is not.  A good early study that supports this point of view is that of Bayoh et al. 2014.  The authors may wish to cite it.  The full citation is:  Bayoh MN, Walker ED, Kosgei J, Ombok M, Olang GB, Githeko AK, Killeen GF, Otieno P, Kariuki S, Desai M, Lobo NF, Vulule JM, Hamel MJ, Gimnig JE. 2014. Persistently high estimates of late night, indoor exposure to malaria vectors despite high coverage of insecticide treated nets.  Parasites and Vectors 7: 380.

1.  In several areas of the text LLINs is mis-spelled as LINN.  Please find these mis-spellings and correct them.

2.  One possible explanation for the relatively high feeding rate on humans as shown by the resting/knockdown collections made indoors is that net integrity has declined, leaving holes open that allows female Anopheles to enter and feed.  If those females are sufficiently resistant (read: less susceptible) to the pyrethroid insecticides or if the potency of the insecticide has worn off over time, then toxicity is lessened and blood feeding facilitated by worn nets.  Aside from references 6 and 7, other studies have suggested this such as Ochomo et al. (2013), which the authors may wish to reference.  Full citation is: Ochomo EO, Bayoh MN, Walker ED, Abongo BA, Ombok MO, Ouma C, Githeko AK, Vulule JM, Yan G, Gimnig JE.  2013. The efficacy of long lasting nets with declining physical integrity may be compromised in areas with high levels of pyrethroid resistance.  Malaria Journal 12: 368.  

3.  Although the use of PCR to detect malaria parasites in Anopheles "heads" has become an accepted practice, the estimation of IBR or EIR with such data is suspect because PCR does not detect infectious sporozoites in salivary glands (which, by the way, are in the thorax).  The authors should add a sentence or explanation about this issue.  Otherwise, the infection rates in the heads are much higher than normally observed with dissection of salivary glands or with sporozoite ELISA.

4.  The references require careful attention to formatting as they are not uniformly formatted.

Overall English language usage is fine.  There are a few places where some edits could be made.

Reviewer 2 Report

This study described data from an entomological survey conducted in coastal and inland Cote D’Ivoire. The result showed a high level of malaria transmission and high proportion of mosquito obtaining human blood and resting indoors despite high LLIN coverage. By comparing two subsets of insects caught from two collection methods (targeting endophagic/endophilic and endophagic/exophilic mosquitoes), the manuscript described mosquito biting behaviors in relation to bed net coverage and usage in the study site. In particular, the result suggested that bed net usage is partially effective at prevent successful blood feeding because a majority of mosquitoes that left the house were unfed. On the other hand, a majority of the ones found indoor had successfully obtained human blood suggesting bed net usage is not fully effective at preventing biting. The study did not aim to understand the underlying reason of the incomplete protection of LLINs nor the difference between the two subsets of mosquitoes but discuss some potential explanations.

Overall, the manuscript is well-written. Importantly, it provides important information and concepts that need to be tackled for future Malaria control and elimination in this region and setting. I have a few minor suggested revisions to help with the clarification of the manuscript.

Overall

Introduction

Line 74-76: Confusing sentence. Please clarify.

Line 92-95: It would be nice if the authors can clearly state the research questions, or hypotheses. There are multiple statistical models being used in data analysis, and often these are driven by specific research questions or hypotheses.

Materials and Methods

Line 116: Are these collections, except the ones in Dec, all done in rainy season? Is there effect of season?

Line 126: Although it’s not very clear to me, I assume these two collection methods were done at different house. And the houses were not reused in subsequent collections. It would be helpful to clearly state the sampling method (e.g., random sampling without replacement?) here. Please clarify if the same houses were used for both collection methods (WT and PSC), and subsequent collections.

Line 127-129: Not clear if the information about bed net usage and coverage come from direct observation or human subject survey-based (i.e., asking people to report their use). Please specify how this information was obtained (e.g., direct observation, use of standardized questionnaire)

Line 140-143: I understand the authors cite detailed methodologies in other publications but for clarity, please at least specify the name of these molecular detection methods. For example, “…species identification using ribosomal DNA PCR-based assay [citation]” or “Plasmodium falciparum detection was done using qPCR-based assay following [citation]”

Results

The authors should consider moving a few tables and figures in Supplementary data section to the main manuscript. Currently, there’s only one table and figure in the main manuscript. For example, Table S1 and S5 and Figure S1 provide interesting and relevant data. Unless there are other technical constraints that I am not aware of, the authors should consider moving them the main text for easy access.

Discussion

Line 290: RT-PCR usually means reverse transcription PCR. Please check if the usage here is correct.

Line 300 and 315: using the word “fraction” can be misleading since the data only show the number, not proportion. Perhaps use “number” instead?

Conclusions

Line 359: The authors mention changes in vector behaviors but what about human behavioral changes that may alter efficacy of LLIN protection? For example, do people spend more time indoor doing activities other than sleeping (so they are outside of bed net but in the house)? If there’s potential for human behavior impact on LLIN protection, this could be another point to discuss in the Discussion section.
